# Physical Activity Capacity Assessment of Patients with Chronic Disease and the Six-Minute Walk Test: A Cross-Sectional Study

**DOI:** 10.3390/healthcare10050758

**Published:** 2022-04-19

**Authors:** Edem Allado, Mathias Poussel, Eliane Albuisson, Jean Paysant, Margaux Temperelli, Oriane Hily, Anthony Moussu, Noura Benhajji, Gerôme C. Gauchard, Bruno Chenuel

**Affiliations:** 1University Center of Sports Medicine and Adapted Physical Activity, CHRU-Nancy, Université de Lorraine, F-54000 Nancy, France; m.poussel@chru-nancy.fr (M.P.); m.temperelli@chru-nancy.fr (M.T.); o.hily@chru-nancy.fr (O.H.); a.moussu@chru-nancy.fr (A.M.); b.chenuel@chru-nancy.fr (B.C.); 2DevAH, Université de Lorraine, F-54000 Nancy, France; j.paysant@chru-nancy.fr (J.P.); gerome.gauchard@univ-lorraine.fr (G.C.G.); 3OMEOS, F-54000 Nancy, France; noura.benhajji@gmail.com; 4Direction de la Recherche Clinique et de l’Innovation, CHRU-Nancy, F-54000 Nancy, France; e.albuisson@chru-nancy.fr; 5IECL, CNRS, Université de Lorraine, F-54000 Nancy, France; 6Département du Grand Est de Recherche en Soins Primaires (DEGERESP), Faculté de Médecine, Université de Lorraine, F-54000 Nancy, France; 7Rehabilitation Medicine Department, CHRU-Nancy, Université de Lorraine, F-54000 Nancy, France

**Keywords:** exercise, sedentary behavior, physical conditioning, human, chronic disease, walk test

## Abstract

Background: This study aimed to evaluate the efficacy of the Six-Minute Walk Test (6MWT) to determine the physical activity capacities of patients with chronic disease. Methods: For this cross-sectional study, we investigated 156 patients with chronic disease and no beta-blocker treatment. They systematically performed a maximal cardiopulmonary exercise test to determine their heart rate peak (HRPeak) and maximal oxygen uptake (V’O_2_max). We considered two groups of patients based upon the results of the functional evaluation of exercise performance: (1) No limitation in exercise performance (V’O_2_max greater or equal to 80% of the theoretical reference) and (2) limited exercise performance (V’O_2_max less than 80% of the theoretical value). All patients also received a 6MWT on the same day as the exercise test. Results: We found 68 (43.6%) patients with normal exercise capacities and 88 (56.4%) patients with limited exercise performance. In this sample, 6MWT mean distances were 510 (87) and 506 (86) m, respectively. There were no significant differences between the two groups for distance and end-test heart rate. The correlation between matrix V’O_2_max measured during the maximal incremental exercise test and the 6MWT distance displayed a positive slope (r = 0.549 CI95 [0.431–0.656]—*p* < 0.001). Conclusion: Our results showed a moderate relationship between 6MWT and physical activity capacity for patients with chronic disease.

## 1. Introduction

Many chronic diseases and more than two million deaths worldwide are attributable to physical inactivity each year. In 2013, the global impact of a sedentary lifestyle was estimated to be USD 67.5 billion per year, divided between USD 13.7 billion in lost productivity and USD 53.8 billion in direct healthcare expenditure—57% of which was borne by the public sector [1]. The benefits of physical activity on chronic diseases have been highlighted in many studies [2,3].

Since 2016, the French Health Ministry has authorized the medical prescription of adapted physical activity (APA) for patients with chronic disease [4]. This activity must be adapted to the pathology and the requirements of the patient, as well as their physical capacities. In addition, an assessment of cardiovascular risk and aerobic exercise capacity is essential to diagnose potential contraindications and to determine a safe level of adapted exercise [5,6]. 

For APA, it is not recommended that a high-level intensity exercise test alone be used to assess sedentary patients with a moderate to high cardiovascular risk. Prior to any medical prescription, the French health authority (HAS) recommends a medical consultation in order to collect data on factors such as anthropometry, as well as the potential pros and cons of practicing an APA [6]. In a second time, an evaluation of cardiorespiratory and muscular capacity must be performed. In order to do so, tests such as the 6-Minute Walk Test (6MWT) have been proposed [6]. However, the real interest in performing such an assessment of exercise capacity in patients suffering from chronic disease that require an APA program needs to be clarified.

In the literature, the 6MWT has been shown to be useful in assessing the aerobic capacity of patients with cardiac or respiratory pathology by showing its connection with maximal oxygen uptake (V’O_2_max) [7,8,9,10]. The 6MWT has also been shown to be effective in assessing this capacity in the elderly, the obese, cancer patients, and in sick children [11,12,13,14,15,16,17,18,19,20]. This test presents the advantage of being standardized, well-documented, and easy to perform at a low budget [21,22].

Thus, we wanted to identify if the 6MWT was suitable for evaluating the aerobic capacities of patients requiring APA who have a chronic disease. Our main goal was to evaluate the level of the relationship between aerobic exercise performance capacity and 6MWT in a sample of chronic disease patients following a hospital-based APA.

## 2. Materials and Methods

This is a cross-sectional, observational, monocentric study performed at the Nancy University Center of Sports Medicine and APA, France, between February 2020 and July 2021 in patients requiring implementation of APA. APA is a means by which the mobility of people who, due to their physical, mental, or social status, cannot perform physical activity under normal conditions, can be improved. 

### 2.1. Participants 

The inclusion criteria were age greater than 18 years with an indication for performing a cardiopulmonary exercise test in order to participate in supervised APA practice inside and outside the hospital. The exclusion criteria were beta-blocker treatment, temporary inability to perform the cardiopulmonary exercise test and 6MWT in the same day, and joint pain or limitations. All inclusions were made systematically and consistently over time.

### 2.2. Ethics Aspects

All data used were obtained from medical records. No supplementary examination was necessary for patients to meet the inclusion criteria. This study is registered with the Information Technology and Freedoms Commission for the University Hospital of Nancy (IRB number: 2021PI191) and on ClinicalTrials.gov (NCT05146544). This study is registered to the “Ethics Committee of the Nancy University Hospital” (number ref: 336 chaired by Pr Martinet) and was designed in accordance with the general ethical principles outlined in the Declaration of Helsinki. The protocol of this study was approved by the Information Technology and Freedoms Commission. All patients gave their verbal consent for the use of their medical data during the period they received medical care at the University Hospital.

### 2.3. Assessment and Intervention 

Data collected from the sample patients were age, sex, and body mass index (BMI). We classified the types of pathologies behind the APA program into groups: Obesity, Rheumatology, Hematology, and Others. A maximum cardiopulmonary exercise test was carried out on all included patients. 

On the same day, following the cardiopulmonary exercise test, the sample patients were split into two groups [23,24]: (1) No limitation of exercise performance capacities: those with normal maximal oxygen uptake (greater than or equal to 80% of the reference value) and (2) limitation of exercise performance capacities defined by a reduced maximal oxygen uptake (less than 80% of the reference value). We used Wasserman equation for reference values [25,26]. The actual peak HR (HRpeak) was measured during the exercise test. 

-Exercise test procedure: V’O_2_max was determined during a maximal incremental exercise test on a cycle ergometer (eBike, GE Healthcare, Boston, MA, USA). Power output was increased by 10% of the power reference value, increasing every 1 min until participant exhaustion. Respiratory and metabolic variables (minute-ventilation, tidal volume, frequency of breathing, V’O_2_, V’CO_2_) were measured breath-by-breath through a mask connected to a pneumotachograph and O_2_ and CO_2_ analyzers (VyntusTM CPX, Vyaire, Höchberg, Germany). Criteria for the achievement of V’O_2_max were HR >90% of the maximal reference value heart rate (210–age), respiratory exchange ratio ([RER] = V’CO_2_/V’O_2_) > 1.1, and/or V’O_2_ plateau.

As recommended, the 6MWT was measured before the exercise test on the same day. The 6MWT was conducted in a 30 m-long corridor following international recommendations [10]. The HR and the best distance were recorded during the procedure.

### 2.4. Measures to Reduce Sources of Bias 

In order to reduce risks of bias, eligible patients who participated in the tests were included into the study protocol in a systematic manner. All the tests (highest effort capacities and 6 min walk test) were carried out according to our pre-established protocol. Prior to the highest-effort capacity test, a calibration was systematically performed. 

### 2.5. Statistical Analysis 

According to the experience of the Center, we expected to find that 20% of subjects would be receiving beta-blocker treatments and 10% of results would be unusable due to pain or limitations. To increase the power of comparisons or correlations, we expected to obtain proportions of close to 50% each for the group of patients with a normal exercise capacity and for the group of patients with limited exercise performance amongst the pool of patients eligible for the study. Using the Simple Asymptotic Method, with a two-sided 95% confidence interval of 16% width (lower limit for proportion 42%; upper limit for proportion 58%), the sample size was 151 subjects. With the 30% of patients not included in the study, the number of participants initially considered was about 200 subjects (calculation = 197). 

Both descriptive and comparative analyses were conducted by accounting for the nature and distribution of the variables. Qualitative variables were described as frequencies and percentages; quantitative variables were evaluated with the mean ± standard deviation (SD) or with the median and interquartile range (IQR). The chi-square test was used for the ordinal or nominal data analysis. We used the student *t*-test to compare age, BMI, HR, and 6MWT results. A binary logistic regression analysis was performed for multivariate analysis of exercise capacities for the estimation of potential confounding factors. We used Pearson’s correlation to analyze the relationship between 6MWT and V’O_2_max. A Bland–Altman plot was applied on standardized values (=[X − mean]/standard deviations) to assess the agreement between the two variables (V’O_2_max and 6MWT). 

The significance level was set at 0.05 for the entire study. IBM™ SPSS Statistics v23 was used for the data analysis.

## 3. Results

### 3.1. Demographic, Clinical Data, and Intervention 

During the 18-month study period, 208 patients were referred to the Nancy University Center of Sports Medicine and APA. Fifty-two patients were excluded from the analysis due to beta-blocker treatment (*n* = 25) or as the cardiopulmonary exercise test and 6MWT were not possible on the same day (*n* = 27) (Appendix A).

One hundred fifty-six patients were included in the study, 122 (78.2%) of whom were female. The mean age was 44.0 (±12.0) years, and the mean BMI was 36.7 (±10.4) kg/m^2^. A total of 68 patients (43.6%) had a normal exercise capacity, while 88 patients (56.4%) only showed limited exercise performance. The range of age and BMI was equivalent across exercise capacity categories (Appendix A). There were no significant differences between the two groups for demographic and clinical characteristics, except for gender, in the multivariate analysis (*p* = 0.008; Table 1). 

### 3.2. Relationship between Physical Activity Capacity and Six-Minute Walk Distance (6MWT)

The correlation between matrix V’O_2_max measured during the exercise test and 6MWT distance showed a positive slope in Figure 1 (r = 0.549 CI95 [0.431–0.656]—*p* < 0.001). In this sample, for normal and limited exercise patients, 6MWT mean distance was 510 (87) and 506 (86) m, respectively. Mean end-test HR was 93.3 (15.3) bpm and 93.9 (14.3) bpm, respectively. There were no significant differences between the two groups for distance and end-test HR (*p* = 0.777 and *p* = 0.815).

A Bland–Altman plot showed an excellent agreement between the two variables (Figure 2), since 146/156 patients (93.6%) showed a mean difference between the two variables within −1.96 SD and +1.96 SD. 

Six patients were above the +1.96 SD line and thus had greater intensity capacities in V’O_2_max than in the 6MWT. Patient characteristics are detailed in Table 2. Amongst these patients, there were as many people with obesity as people with a normal BMI (*n* = 3); four women and two men were included. Five out of six of these patients had normal exercise capacity.

Four patients were below the −1.96 SD line and thus had considerably higher results in the 6MWT (Table 2). Only one patient presented with obesity—most of the patients came from the oncology department. It is important to note that all of the patients in the group had limited exercise capacity.

## 4. Discussion 

The study described a moderate correlation between 6MWT and V’O_2_max, but an excellent agreement using Bland–Altman plotting of the two tests for patients with chronic disease requiring APA. However, the 6MWT does not allow a precise determination of the patients’ level of exercise capacity, and a maximal exercise challenge should still be recommended in patients that need such degree of precision (search for exercise contraindications, noticeably debilitated patients, etc.) prior to APA.

Although the patients included in our study had a greater exercise capacity than patients with COPD or heart failure, as described in the literature [27,28,29], the relationship level found the same size effect as that observed in previous studies in patients with cardiac or respiratory disease or obesity [8,9,15,17,30,31]. In fact, a recent meta-analysis has shown that the correlation of V’O_2_max and the travelled distance was better correlated for patients with COPD (r = 0.65, 95% CI 0.61–0.70) [32]. Supporting this observation, the correlation was even greater while performing the incremental shuttle walk test (r = 0.81, 95% CI 0.74–0.85) [32].

Even though our study highlighted a 6 m average difference between patients with a low versus normal capacity (510 (87) m and 506 (86) m). No statistically nor clinically significant difference was pointed out (thresholds 14.0 to 30.5 m may be clinically significant across multiple patient groups) [12]. We concluded that the 6MWT is not adapted to discriminate patients with low versus normal capacity. 

However, although the measurement of V’O_2_max during a test on a cycle ergometer is the reference for estimating a patient’s exercise capacity [33], it can be limiting for patients presenting with chronic conditions, such as those with breast cancer [13]. Moreover, in our study, we observed that amongst four patients who presented the highest effort capacities at the 6-Minute Walk Test versus the V’O_2_max test, 50% had a cancer (n°7 and n°8 in Table 2). This observation is in agreement with the previously described limitations of the V’O_2_max [13]. 

Therefore, the 6-Minute Walk Test is a good method for evaluating the effort capacity of a patient. In fact, the walk test presents the advantage of being easy to perform and convenient, as it does not require any specific equipment for its realization and its interpretation [10]. Nevertheless, even though we saw a moderate correlation, as seen for the 1-Minute Sit-to-Stand Test, the 6MWT does not discriminate between the two capacity levels observed [8,34]. 

This study has limitations: Firstly, the size sample, which was taken from a single study in one hospital center, prevents any extension of the results. Additionally, the results of the study are only applicable to adults and the presence of many obese participants limited a subgroup exploration of different chronic conditions.

The main strength of our study is the use of V’O_2_max as a gold standard for assessing the functional capacities of patients. This study also incorporates all the recommendations of the French Health Act issued by the Ministry of Social Affairs and Health.

## 5. Conclusions

In conclusion, our study found that there was only a modest relationship between 6MWT distance and V’O_2_max. The 6MWT did not succeed in delineating the aerobic capacity of patients with chronic disease that require APA. Further interventional and controlled studies are needed to explore the efficiency of the 6MWT or incremental shuttle walk test in evaluating the exercise capacity of patients requiring APA in a larger cohort of patients. 

## Figures and Tables

**Figure 1 healthcare-10-00758-f001:**
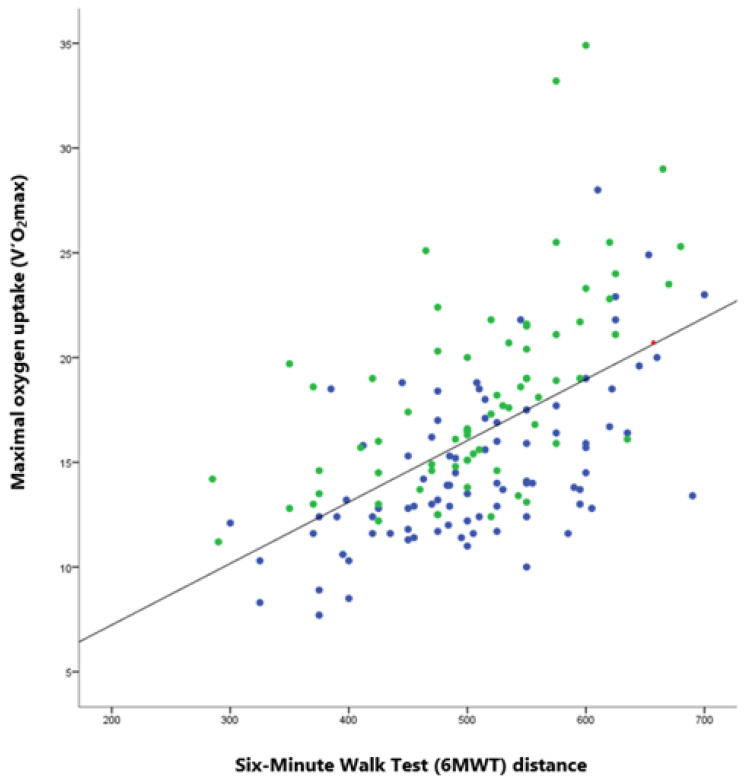
The relationship between matrix maximal oxygen uptake (V’O_2_max) measured during the exercise test and the Six-Minute Walk Test (6MWT) distance by level exercise capacity (linear regression modelling). Legend: Green points for normal exercise capacities: Maximal result of exercise test is greater or equal to 80% of the reference value. Blue points for lower exercise capacities: Maximal result of exercise test is less than 80% of the reference value.

**Figure 2 healthcare-10-00758-f002:**
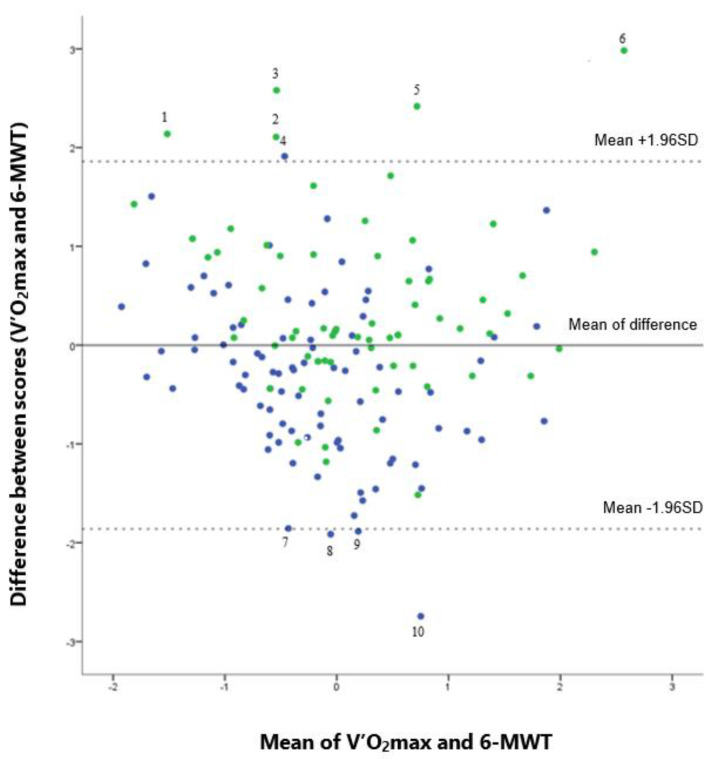
Bland and Altman plot showing the agreement between the V’O_2_max and the Six-Minute Walk Test (6MWT) distance. Legend: Green points for normal exercise capacities: Maximal result of exercise test is greater or equal to 80% of the reference value. Blue points for lower exercise capacities: Maximal result of exercise test is less than 80% of the reference value.

**Table 1 healthcare-10-00758-t001:** Baseline demographic and clinical characteristics (*n* = 156).

	Normal Exercise Capacity (*n* = 68)	Lower Exercise Capacity (*n* = 88)	* *p*-Value	
			Univariate	Multivariate
Female	60 (88.2)	62 (70.5)	**0.006**	**0.008**
Male	8 (11.8)	26 (29.5)		
Age, years	45.5 (12.0)	42.9 (12.0)	0.179	0.075
Body mass index, kg/m^2^	37.4 (9.8)	36.3 (10.8)	0.505	0.334
Disease categories - Obesity - Rheumatology - Hematology and Oncology - Nephrology - Diabetology - Pneumology - Internal medicine - Other	42 (61.8) 9 (13.2) 4 (5.9) 1 (1.5) 1 (1.5) 1 (1.5) 4 (5.9) 6 (8.8)	54 (61.4) 12 (13.6) 2 (2.3) 1 (1.1) 0 (0.0) 2 (2.3) 6 (6.8) 11 (12.5)	0.857	0.909

Legend: Data are presented as *n* (%) for dichotomous variables, mean (SD) for continuous demographic variables with normal distribution and median [interquartile range] with non-normal distribution. * We use the student *t*-test to compare age and body mass index, and the chi-square test or Fisher’s exact test for other variables.

**Table 2 healthcare-10-00758-t002:** Characteristics of the 10 patients who exceeded the upper (A) and lower (B) agreement limit in the Bland–Altman plot (Figure 2) and thus have considerably higher maximal oxygen uptake (V’O_2_max) than 6-Minute Walk Test (6MWT) distance.

	Patient ID Number in Figure 2	Sex/Age/BMI	Department	6MWT Distance	Peak Cycle Work Rate *	V’O_2_max *	Hrpeak *
**Upper agreement** **Limit (A)**	**1**	♀ 64 years 41.7 kg/m^2^	Obesity	285 m	80 W (108%)	14.2 mL/min/kg (101%)	134 bpm (86%)
	**2**	♂ 59 years 37.8 kg/m^2^	Obesity	370 m	105 W (74%)	18.6 mL/min/kg (87%)	160 bpm (99%)
	**3**	♀ 44 years 37.6 kg/m^2^	Obesity	575 m	120 W (111%)	15.9 mL/min/kg (86%)	160 bpm (91%)
	**4**	♀ 57 years 23.3 kg/m^2^	Rheumatology	385 m	70 W (86%)	18.5 mL/min/kg (79%)	125 bpm (77%)
	**5**	♀ 36 years 22.4 kg/m^2^	Rheumatology	465 m	130 W (104%)	25.1 mL/min/kg (89%)	164 bpm (89%)
	**6**	♂ 37 years 23.8 kg/m^2^	Rheumatology	600 m	220 W (96%)	34.9 mL/min/kg (92%)	187 bpm (102%)
	**N°**	**Sex/Age/BMI**	**Department**	**6MWT** **Distance**	**Peak Cycle Work Rate ***	**V’O_2_max ***	**HRpeak ***
**Lower agreement limit (B)**	**7**	♀ 55 years 20.2 kg/m^2^	Oncology	550 m	80 W (91%)	10.0 mL/min/kg (40%)	139 bpm (84%)
	**8**	♂ 67 years 27.1 kg/m^2^	Oncology	585 m	60 W (43%)	11.6 mL/min/kg (47%)	103 bpm (67%)
	**9**	♂ 57 years 28.4 kg/m^2^	Rheumatology	605 m	75 W (47%)	12.8 mL/min/kg (49%)	146 bpm (90%)
	**10**	♀ 42 years 40.4 kg/m^2^	Obesity	690 m	80 (72%)	13.4 mL/min/kg (74%)	169 bpm (95%)

* Measurements and percentages of the reference value for peak cycle work rate, V’O_2_max, or HRpeak. Legend: The 10 patients were identified after performing the Bland–Altman plot. The Patient ID Numbers are those identified in Figure 2.

## Data Availability

Not applicable.

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
