# Peer review of "Physical Activity Capacity Assessment of Patients with Chronic Disease and the Six-Minute Walk Test: A Cross-Sectional Study"

_healthcare, 2022, doi:10.3390/healthcare10050758_

Round 1
Reviewer 1 Report
The authors of the article had the following objective: To evaluate the efficacy of the Six-Minute Walk Test (6MWT) to determine the physical activity capacities of patients with chronic diseases.
My comments and suggested changes to the article are as follows:
- They could provide a STROBE checklist for observational studies.
- Enter a minimum of 5 keywords defining your research.
- The authors do not explain the scientific rationale and basis of the reported research, the objective of evaluating the efficacy of the Six-Minute Walk Test 17 (6MWT) should be presented with a broad justification on the subject since the 6-minute walk test (6MWT) is a safe, standardized and well-used method for assessing functional capacity in various populations with pathology.
- Describe the relevant setting, locations and dates, including recruitment, exposure, follow-up and data collection periods.
- Explain how the sample size was determined.
- Explain the measures taken to address potential sources of bias.
- Enter a flow chart.
- The VOâ‚‚ max is the maximum amount of oxygen that the organism can absorb, transport and consume in a given time, you indicate that you took patients from 18 years forward, don't you think it may be a big bias or limitation to compare young people with elderly people?
- Discussions should cover the key findings of the study: discuss any previous research related to the topic to place the novelty of the finding in the appropriate context, discuss possible shortcomings and limitations in their interpretations, discuss their integration into the current understanding of the problem and how. This advances current views, speculates on the future direction of the research, and freely postulates theories that could be tested in the future, completed and reformulated, broaden their discussion, broadly defining their limitations.
- The bibliographic references used are scarce and very old, for example, reference 7 (1996), reference 8 (1985), reference 14 (1996), reference 16 (1990), practically all the references are more than 20 years old, which indicates an unoriginal and uninteresting research for the readers.
- The same bibliographic references are used to write and interpret the introduction and discussion of this research article, see references 6,7,8,10,11,13.
Reviewer 2 Report
I thank the authors for submiting their manuscript. Here he left some comments and suggestions:
- First of all, there is an important limitation that corresponds to the design of study (croos-sectional), however, I believe that the authors can improve the presentation of their manuscript.
- I suggest ordening the methodology in: design, participants (inclusion/exclusion criteria), ethics aspects, assessments, intervention and statistican analysis.
- Change: gener x sex; women x female; men x male.
- In the results, the presentation of Tables 2 and 3 is not entirely clear, why not carry out an analysis considering the distribution of sex, pathologies or others?
- The location of figure 1 and tables 2 and 3 must be in results!
- In think the discussion can be strengthened and include the practical implications of the study.
Reviewer 3 Report
My recommendations are:
For keywords, I recommend deleting the 6-MWT line.
I recommend extending the Introduction section with new relevant aspects of the study.
Lines 52-52 repeat the idea of ​​the purpose mentioned below, I recommend clarification.
Bibliographic indexes in the text must be entered before the full stop, I recommend correction.
Lines 60-63 refer to the inclusion criteria in the first session, I recommend that you mention whether these criteria have been further monitored. The study was conducted over a period of more than one year, when the testing was actually performed, I recommend further clarifications.
I recommend that section 3.1 be further moved to section 2.1. It is a presentation of the sample, not the results themselves.
Table 1 shows only the data on the group of women, which represent 122 people. Men are missing, I recommend correction.
Lines 65-68 mentioned that you divided your subjects into four categories of diseases, and according to table 1 there are 8 categories. I recommend clarification.
The study covers a period of more than 1 year, the relevance of the time in relation to the intervention is not clear.
I recommend extending the disc section.
It is very difficult to trace the logic behind the design of this article. Also, the novelty comes as a finding and not as a concrete intervention. The results are not well specified, mention only at the end of the article the results that exceed the limits. The other results on the two groups are missing.
In tables 2 and 3 mention the Clinical characteristics column and mention the department, I recommend clarification. Position these tables at the end of the article, after the conclusions, as annexes ... I recommend clarification.
I recommend that the results be clear on the two samples.
With 20 bibliographic clues, this article is not sufficiently grounded in theory. The most recent bibliographic index is from 2018, I think that in the last 4 years other material has certainly been written that could be relevant.
The title is not relevant to the content, I recommend reformulating. What is the interest, did not emerge from the content.
In conclusion, this study does not have a good logic and the way of editing has big gaps. The level of this article is not for such a relevant journal in the field.
Round 2
Reviewer 1 Report
As I indicated above, the article has major limitations and possible risks of bias, which have not been corrected, and the following are the aspects that remain to be corrected:
1º The STROBE checklist is not provided as indicated.
2º They still do not specify on what they based the calculation of the sample size to determine any of the conclusions, then in their limitations they indicate that the sample size is small but we do not know how they have calculated it or on what basis they establish as correct the number of participants they select.
3º They do not explain the measures adopted to address possible sources of bias.
4º They have not introduced the flow chart as instructed.
5º There were significant differences between the two groups in terms of sex when this is a determinant of VO2 max.
The obesity of the participants, which affects half of the participating subjects, is also a conditioning factor which has not been taken into account at the time of VO2 max.
But what puzzles me most is how you indicate in your Table 1 a mean age of 45.5 (12.0) for the normal exercise capacities group and for the lower exercise capacities group 42.9 (12.0), but in Table 2 patients 1 are 64 years old, patients 2 are 59 years old, patients 8 are 67 years old.
Therefore, this does not correspond to the data in their Table 1, which indicates many doubts as to the reliability and validity of the data.
6º It was previously indicated that both your introduction and discussion presented ambiguous and irrelevant bibliography, you should reformulate all its content since it is not current and does not present an update and detailed justification of the subject of study, I again indicate some references more than 20 years old: reference 8 (1996); reference 9 (1985); reference 10 (2003); reference 11 (2001); reference 12 (2004); reference 13 (1999); reference 14 (2003); reference 15 (1996); reference 24 (1991); ....
Reviewer 2 Report
I thank the authors for the new version, wich is much clearer.
Reviewer 3 Report
I recommend extending the introductory section with a focus on issues related to the purpose of the study.
Extending the Discussions section, compared to the indexes mentioned in the previous sections, to add only 3 new indexes.
I recommend that in table 1 you detail by sex: age, BMI and diseases.
I recommend that you specify how the Sex / Age / BMI categories were divided and the number of subjects in Table 2.
